# Comparison of Surgical Outcomes between Robotic Transaxillary and Conventional Open Thyroidectomy in Pediatric Thyroid Cancer

**DOI:** 10.3390/cancers13133293

**Published:** 2021-06-30

**Authors:** In A Lee, Kwangsoon Kim, Jin Kyong Kim, Sang-Wook Kang, Jandee Lee, Jong Ju Jeong, Kee-Hyun Nam, Woong Youn Chung

**Affiliations:** 1Department of Surgery, Yonsei University College of Medicine, Severance Hospital, Seoul 03722, Korea; anzelina@yuhs.ac (I.A.L.); JKKIM3986@yuhs.ac (J.K.K.); JANDEE@yuhs.ac (J.D.L.); JUNGJONGJ@yuhs.ac (J.J.J.); KHNAM@yuhs.ac (K.-H.N.); WOUNGYOUNC@yuhs.ac (W.Y.C.); 2Department of Surgery, College of Medicine, The Catholic University of Korea, Seoul 06591, Korea; noar99@naver.com

**Keywords:** pediatric thyroid cancer, robotic trans-axillary thyroidectomy, conventional open thyroidectomy

## Abstract

**Simple Summary:**

Thyroid cancer is rare in the pediatric population, but in comparison to thyroid carcinomas in adults, those occurring in children present with aggressive and advanced features. Recent innovations in the surgical technique of robotic thyroidectomy for young patients have offered the opportunity to improve cosmetic effects and oncologic outcomes. The aim of our retrospective study was to conduct a review of the pediatric population who underwent robotic trans-axillary thyroidectomy or conventional open thyroidectomy between February 2008 and December 2019. In the hands of an experienced surgeon, robotic thyroidectomy is a feasible and safe option for pediatric patients.

**Abstract:**

Thyroid cancer in children is very uncommon. For pediatric thyroid cancer, robotic surgery has served as a minimally invasive surgical alternative to conventional open surgery. Our study aimed to evaluate the results of robotic versus open surgical treatment for patients with thyroid cancer younger than 20 years of age at the time of diagnosis. This retrospective review included 161 pediatric patients who underwent robotic transaxillary or conventional open thyroidectomy at our institution from 2008 to 2019. Of these patients, 99 comprised the robotic group and 62 the open group. Patient demographics, surgical outcomes, and disease-free survival rates were compared between the two groups. Patients in the open group were more likely to have advanced stage diseases with a larger tumor size and higher tumor-node-metastasis stage than those in the robotic group. Operation time and follow-up period were similar in both groups. Patients in the robotic group had a lower rate of postoperative complications and a shorter length of hospital stay, but they also had a lower average number of retrieved central lymph nodes. However, there were no significant between group differences in recurrence rates and disease-free survival. In the hands of an experienced surgeon, robotic thyroidectomy is a feasible and safe option for pediatric patients.

## 1. Introduction

Thyroid cancer was previously uncommon in the pediatric population. However, the incidence of thyroid cancer in children has rapidly continued to increase in the last three decades. Thyroid cancer in advanced stages with local or distant metastasis at the time of diagnosis and recurrence is more common in children than in adults, but the prognosis remains very favorable [1,2,3,4,5,6]. The important clinical, molecular, and pathological differences in differentiated thyroid cancer among children compared to that in adults have prompted the development of unique pediatric guidelines. The American Thyroid Association (ATA) published guidelines for pediatric thyroid cancer in 2015 in consideration of the underlying biologic and molecular differences between pediatric and adult thyroid cancer [2,7]. The treatment of pediatric thyroid cancer generally involves surgery and postoperative treatment, involving radioactive iodine therapy and thyroid-stimulating hormone suppression. Surgery is the main approach for treating pediatric thyroid cancer. The extent of surgery ranges from lobectomy to bilateral total thyroidectomy (BTT). According to the guideline, owing to the increasing incidence of bilateral and multifocal disease, in the majority of children, BTT with or without central compartment node dissection (CCND) is recommended. Since locoregional or distant metastasis is more likely to be present in young patients, more aggressive treatment is recommended for children than for adults [7,8].

Owing to the small body size of the pediatric population, open cervical approaches were performed in the past. However, in consideration of the impact on the quality of life and the development of growing pediatric patients, minimally invasive approaches have been preferred recently [9,10,11]. Robotic transaxillary thyroidectomy (RT) for pediatric patients was first reported by Lobe et al. in 2005 [12]. Since then, only a few studies have reported on the applicability of robotic surgery for pediatric patients with thyroid cancer, although there are already many reports on the advantages of RT for adults with large patient series showing similar oncologic outcomes and morbidity and better cosmetic results than conventional open thyroidectomy (COT) [13,14,15,16,17].

Therefore, the aim of this study was to retrospectively analyze surgical outcomes in pediatric patients with thyroid cancer who underwent RT or COT. 

## 2. Patients and Methods

### 2.1. Patients

This study evaluated pediatric patients (age ≤ 19 years) at the time of diagnosis who underwent RT or COT for thyroid cancer at Severance Hospital, Yonsei University College of Medicine, Seoul, Korea, between February 2008 and December 2019. Clinical and pathological data were collected retrospectively and stored in a dedicated database for analysis. At our institution, eight surgeons were involved in this study. Patients who were diagnosed with thyroid cancer in the outpatient clinic were routinely provided explanations of the operative techniques involved in COT and RT, and the patients themselves or their parents chose their preferred surgical procedure. Six surgeons were experts on RT, and two doctors performed only COT. In some high advanced cases, COT was recommended as per the surgeon’s opinion. A total of 161 patients were enrolled, excluding patients who visited our hospital due to recurrence after a primary thyroid cancer operation performed at another hospital. Of these patients, 99 underwent RT and 62 underwent COT. The study protocol was approved by our institutional review board (IRB: 4-2021-0470). The institutional review board waived the requirement for patient consent. 

All the patients were preoperatively diagnosed by means of physical examination, ultrasonography (US), and neck computed tomography (CT), and the diagnosis was histologically confirmed by US-guided fine-needle aspiration (FNA). Previously, the evaluation, treatment, and follow-up of children with thyroid cancer had been performed following adult guidelines, until the ATA was prompted to specifically address treatment for children with benign and malignant thyroid tumors [7,18,19].

### 2.2. Operative Procedure

RT was performed using the da Vinci Si, Xi, or SP surgical robotic system (Intuitive Surgical, Sunnyvale, CA, USA) via a gasless, transaxillary approach, as previously published in a serial report from our institution [14,17,20,21]. In brief, the patient was placed in a supine position with the neck slightly extended. The arm was raised to allow the shortest distance between the axilla and the anterior neck. A 5–6 cm vertical incision was made in the axilla, and a subplatysmal skin flap was dissected from the axilla to the anterior neck, through the sternocleidomastoid (SCM) muscle bifurcation. Through the avascular space between the sternal and clavicular heads of the SCM muscle, the strap muscles were elevated from the thyroid gland until the contralateral thyroid gland was exposed. To maintain a working space, a spatula-shaped external retractor (Chung’s thyroid retractor) was inserted through the axillary skin incision. Apart from docking of the robotic arms, including all three instruments and the camera, all the RT procedures were similar to the COT procedures. 

### 2.3. Postoperative Follow-Up

The following clinical parameters were analyzed: patient characteristics, surgical variables, extent of surgery, pathological findings, and postoperative outcomes including the recurrence rate. Pathologic examinations included assessments of the type of cancer, tumor size and number, gross extrathyroidal extension (ETE), disease tumor-node-metastasis (TNM) stage, number of lymph nodes (LNs) harvested, and number of metastatic LNs. We also assessed perioperative complications, including hematoma, seroma, wound infection, hypocalcemia (transient or permanent), chyle leakage, vocal cord palsy (transient hoarseness or permanent recurrent laryngeal nerve palsy), oculo-sympathetic paresis (Horner’s syndrome), trachea injury, esophageal injury, and brachial plexus neuropraxia. All the patients were followed up on in the same manner, including clinical examinations within 1 week of discharge and a 3- to 6-month follow-up that included a physical examination, neck US, or CT and an assay of tumor markers (serum thyroglobulin concentration).

### 2.4. Statistical Analysis

All statistical analyses were performed using the Statistical Package for the Social Science software for Windows version 24.0 (IBM Corp., Armonk, NY, USA). Continuous quantitative data are expressed as means ± standard deviations (SDs) and categorical qualitative data as percentages. Data from the two patient groups were statistically compared using the chi-square test or the Mann–Whitney U test as appropriate. Kaplan–Meier survival analysis was used to analyze survival rates according to each variable. All *p* values less than 0.05 were considered statistically significant.

## 3. Results

### 3.1. Demographic and Clinical Data

Among a total of 161 patients (22 male and 139 female patients), 99 patients (6 male and 93 female; sex ratio 1:15.5) underwent RT, and 62 patients (16 male and 46 female; sex ratio 1:2.9) underwent COT. The proportion of female patients was significantly higher in the robotic than in the open group (*p* < 0.001). The mean age and body mass index (BMI) were equivalent between the robotic and open groups. Incidentaloma was the most common reason for patient visits in the robot and open groups (82.8% and 67.7%, respectively, *p* = 0.317), and nonspecific symptoms included hypothyroidism or hyperthyroidism, bloody sputum, headache, short stature, neck swelling, and neck pain. There was no history of radiation exposure in either group and no significant difference in family history (*p* = 0.700). The results for the clinical characteristics of the two groups are presented in Table 1.

### 3.2. Surgical Outcomes and Pathologic Findings

Table 2 shows the operation type and intraoperative outcomes between robotic and open groups. The extent of thyroidectomy was classified as lobectomy, ipsilateral total thyroidectomy with contralateral partial or subtotal thyroidectomy, and BTT, and each accounted for 31.3%, 14.1%, and 53.5% of the operations in the robotic groups and 17.7%, 4.8%, and 77.4% of the operation in the open groups, respectively (*p* = 0.008). There was a significant difference in the extent of node dissection between patients who underwent CCND plus BTT only and those who underwent modified radical neck dissection (MRND) plus BTT in the two groups (*p* = 0.01). The mean operation time in the robotic group was 171.2 ± 101.7 min (console time, 76.9 ± 52.0, including only the 71 most recent cases, due to the lack of a recording system) compared with 182.6 ± 98.2 min in the open group; this difference was not statistically significant (*p* = 0.496). The combined procedure during operation for locally advanced cases, such as recurrent laryngeal nerve (RLN) or tracheal shaving or reanastomosis, detected three (3.0%) cases in the robotic group, and seven (11.3%) cases in the open group (*p* = 0.035).

The pathologic findings with the TNM stage are shown in Table 3. There were no significant differences between the two groups in type of tumor, tumor number (multifocality or bilaterality), or gross ETE. However, the tumor size was significantly larger in the open than in the robotic group (2.5 ± 1.8 vs. 1.8 ± 1.2, respectively, *p* = 0.01). The mean number of dissected central LNs was 7.3 ± 5.1 in the robotic group and 11.3 ± 7.0 in the open group (*p* < 0.001), with central LN metastasis observed in 2.8 ± 3.2 in the robotic group and 6.0 ± 5.2 in the open group (*p* < 0.001). However, the mean number of harvested LNs in cases of MRND was not significantly different between the two groups (*p* = 0.107). The mean number of central and lateral neck LN metastasis was 6.2 ± 4.1 in the robotic group and 12.5 ± 11.2 in the open group (*p* = 0.012).

Comparison of the TNM stages according to the American Joint Committee on Cancer (AJCC) 8th edition guidelines revealed that there were significant differences in the T, N, and M stages (*p* = 0.01, *p* = 0.001, and *p* = 0.009, respectively) between the two groups. There was one case of metastatic papillary carcinoma detected in the central LNs without any thyroidal lesion in pathological findings of RT, classified as T0 in Table 4. When subdivided, the proportion of T0 to T2 stage did not differ significantly between the two groups (*p* = 0.401); however, the frequency of T3 to T4 stage was significantly higher in the open group (*p* = 0.013). Advanced N1 (*p* = 0.001) and M1 stage (*p* = 0.009) were observed more frequently in the open group. The routine gene mutation test in our institution was conducted for the BRAF mutation from 2015 and TERT mutation from 2019; therefore, the number of cases was relatively small, but there was no significant difference in positive for the BRAF mutation in both groups (21% in the robotic and 18% in the open group, *p* = 0.315). TERT mutation status was confirmed in a total of 10 patients in both groups (four patients in the robotic and six patients in the open group), and there were no mutations in any of the patients (Table 4) [22].

Complications after surgery occurred in 21 (21.2%) cases in the robotic and 25 (40.3%) cases in the open group (*p* = 0.009) (Table 5). The postoperative complications observed included transient or permanent hypocalcemia, chyle leakage, wound infection, transient or permanent RLN injury, and oculo-sympathetic paresis (Horner’s syndrome). Transient hypocalcemia was the most common complication in both groups (robotic: 15 (15.2%) patients, open: 18 (29.0%) patients, *p* = 0.076). In the robotic group, permanent hypocalcemia and chyle leakage were observed in two (2.0% for both) patients and wound infection and transient hoarseness in one (1.0% for both) patient, each. In the open group, three (4.8%) cases of chyle leakage, two (3.2%) cases of permanent hypocalcemia, and one (1.6% for both) case of RLN injury and Horner’s syndrome were detected. The length of postoperative hospital stay was significantly shorter in the robotic than in the open group (3.6 ± 1.1 days vs. 4.8 ± 2.0 days, respectively, *p* < 0.001). 

The mean follow-up time did not differ significantly (robotic: 67.0 ± 38.5 (6–147) and open: 72.1 ± 42.7 (8–155), months (range), *p* = 0.569). During the follow-up period, serum thyroglobulin, neck US, and/or neck CT showed recurrence in seven patients in each group (7.1% and 11.3%, respectively, *p* = 0.355). In each group, there were cases of local relapse, distant metastasis, and a combination of local and distant metastasis (3%, 3%, 1% in the robotic, and 8.1%, 1.6%, 1.6% in the open group, respectively, *p* = 0.472) (Table 6). Disease-free survival (DFS) was 92.9% after robotic surgery and 88.7% after open surgery; this difference was not significant (Log-rank *p* = 0.938, Figure 1).

## 4. Discussion

Pediatric thyroid cancer can be differentiated from adult thyroid cancer based on some characteristics, such as a high frequency of ETE, increased multifocality and bilaterality, a high incidence of nodal involvement, a higher risk of recurrence, and an excellent prognosis [2,3]. The ATA guidelines recommend aggressive surgery (subtotal or BTT) for the treatment of pediatric thyroid cancer. However, since massive CCND can lead to serious complications, such as RLN injury or hypoparathyroidism, prophylactic CCND in pediatric patients is recommended to be selectively considered based on tumor focality, tumor size, and experience of the surgeon [7].

Nonetheless, the surgical method must be chosen with care, considering the influence on the quality of life and the development of growing pediatric patients [23]. Moreover, surgical treatment has been shifting toward a conservative approach in patients with low and intermediate risk differentiated thyroid cancer in recent years [24]. Over the last decade, robotic surgical systems have provided surgeons with a three-dimensional working field, a magnified view, a tremor-filtering system, and multiarticulated instruments. The effects of RT on oncologic outcomes and morbidity are equivalent or similar to those of COT; RT leads to improved cosmetic outcome, as well as reduced postoperative discomfort with respect to swallowing difficulty [11,15,17,25].

Due to their physical limitations and aggressive nature of the cancer, pediatric thyroid cancer patients traditionally underwent COT in the past. However, recent advances in technologies for robotic surgery offer pediatric patients an opportunity to choose the surgical method. Even at our institution with high-volume surgeons, of the 161 patients, 99 (61.5%) chose RT. Although thyroid cancer is more common among females than males, the robotic group included a relatively higher proportion of women than the open group (1:15.5 vs. 1:2.9, respectively, *p* < 0.001). This may be because of greater concerns regarding cosmetics in women than in men, and a similar phenomenon was observed in other RT studies [26]. BMI was similar between the two groups (robotic: 22.2 ± 3.9 (14.1–39.7) kg/m^2^ vs. open: 22.0 ± 4.8 (13.2–35.9) kg/m^2^, *p* = 0.425). In previous studies from the United States, RT in pediatric patients was limited by the patient’s BMI, but in our institution, the BMI range of RT was 13.2–35.9 kg/m^2^ [13,27]. Thus, this shows that RT is not affected by the patient’s BMI (underweight to obese).

Upon comparing the pathological findings of RT and COT, the open group was found to have significantly larger average tumor size (*p* = 0.01); more aggressive features of tumor that required shaving or reanastomosis of surrounding nerves, muscle, or tracheal wall (*p* = 0.035); and more frequent occurrences of MRND, due to lateral neck node metastasis (*p* = 0.01), with significantly higher TNM stages (*p* = 0.01, *p* = 0.001, and *p* = 0.009, respectively), than the robotic group. This disproportionate distribution may be explained by the social perception that open surgery is more accurate for highly advanced cancer or the surgeon’s recommendation at the outpatient clinic. Despite this, the average tumor size in the robotic group was 1.8 ± 1.2 cm, which was smaller than that in the open group (2.5 ± 1.8 cm), but RT was possible even for tumors larger than 4 cm (nine cases in total, maximum diameter 5.3 cm). In addition, RT was safe, especially for the preservation of important structures such as the nerves. One case (1%) of hoarseness occurred due to temporary RLN traction injury, but there were no permanent RLN injuries or sympathetic paresis (Horner’s syndrome), despite three advanced cases of T4 in the robotic group. All three cases of RLN shaving in the robotic group had no postoperative complications. To minimize the rate of intraoperative complications, our institution additionally introduced intraoperative neuromonitoring in some cases, such as clinical N1, clinically suspicious T4, and preoperative vocal cord paralysis, from 2015, and no cases of postoperative RLN injury were reported between 2015 and 2018 [28]. Likewise, RT could be performed in children with advanced cancers, such as those with a large tumor of 4 cm or more, gross ETE with RLN or tracheal wall invasion, without postoperative complications. T4 stage, determined according to the 8th edition of the AJCC guideline, had a negative effect on survival rate [29]. The open group had a significantly higher number of patients with advanced T stages; however, there was no statistically significant difference in the recurrence rate or DFS (*p* = 0.355, *p* = 0.938, respectively) between the two groups. 

Papers published 10 years ago reported that the RT procedure requires an average of 30–40 min more than open surgery; however, among pediatric patients, there was no significant difference in operation time between the two groups (*p* = 0.496); in fact, the average operation time was 11.4 min shorter for the robotic group [15,25,26]. This difference may be because of the higher frequency of difficult operations, e.g., more patients with MRND in the open than in the robotic group (33 [53.2%] cases vs. 23 [23.2%] cases, respectively). This also means that the technologies for RT have been advancing rapidly. In addition, the robotic group also had shorter hospital stays after surgery than the open group (*p* < 0.001). 

Overall, RT for pediatric patients demonstrated similar results as COT in terms of surgical outcomes, but the number of harvested LNs by CCND was lower in the robotic group than in the open group (*p* < 0.001), possibly because there were relatively more cases of lobectomy in the robotic group than in the open group (29 [29.3%] cases vs. 9 [14.5%] cases, respectively). Additionally, this observation may have been the result of careful consideration of the impact of excessive CCND, such as RLN injury or hypoparathyroidism, adversely affecting the long-term prognosis of pediatric patients [30]. The lack of a significant difference in the number of retrieved LNs in advanced cases, such as MRND (*p* = 0.107), can be interpreted as indicating that the lower number of harvested LNs by CCND in RT was not due to technical issues. As the guidelines indicating BTT or conservative treatment have been established recently, in our institution, the guidelines for pediatric thyroid cancer patients were established on the basis of the previous ATA guidelines for adults [18]. As mentioned above, prophylactic CCND is selectively recommended for children depending on tumor focality, tumor size, and experience of the surgeon, according to ATA guidelines [7]. However, prophylactic CCND has been practiced by almost all head and neck surgeons, and endocrine surgeons in South Korea and Japan before the release of this guideline [31,32,33]. Therefore, except for five patients (three in the robotic group and two in the open group) who were diagnosed with nondiagnostic (e.g., cell paucity) or indeterminate nodules (Bethesda category I or III) using preoperative FNA specimens, prophylactic CCND was implemented in all cases.

The strength of this study was that it compared surgical outcomes between RT and COT using a large series of pediatric patients. Numerous studies have shown the advantages of RT in adult patients who had excellent outcomes, and these advantages are also applicable to the pediatric population [12,13,14,15,16,17,20,21,25,26,34,35]. First, a three-dimensional working field with a magnified view provided by robotic systems permits visual accuracy for smaller strictures in pediatric patients, and tremor-filtering multiarticulated instruments enable a wide range of surgical treatments involving small incisions and precise, meticulous manipulation of the surgical tissues in children with thyroid cancer. Second, although it is difficult to accurately compare efficiency between the procedures because the number of advanced cancer cases was higher in the open group than in robotic group, RT was associated with a similar operation time, recurrence rate, and DFS as COT and yielded better results with respect to a reduced length of hospital stay and low risk of postoperative complications than COT. Finally, the cosmetic outcome of RT helps alleviate the psychosocial impact of surgery and subsequent scarring in children [11,13,34].

The limitations of our study include a selective bias toward patients with advanced cancers in the COT group; hence, it was difficult to clearly elucidate the superiority of RT to COT for highly advanced thyroid cancer. Additionally, the incidence of thyroid cancer in children has increased rapidly in the last 20 years; thus, more follow-up is required to confirm long-term DFS. Therefore, prospective, multicenter controlled studies with long-term follow-up are needed to determine the operative outcomes and oncologic safety of RT for the pediatric population.5. Conclusions

The incidence of pediatric thyroid cancer has been increasing steadily over the last three decades. RT serves as a minimally invasive surgical alternative to COT. Recent innovations in the surgical technique of RT for young patients offer the opportunity to improve cosmetic and surgical outcomes. In the hands of an experienced surgeon, RT is a feasible and safe option for pediatric patients. 

## Figures and Tables

**Figure 1 cancers-13-03293-f001:**
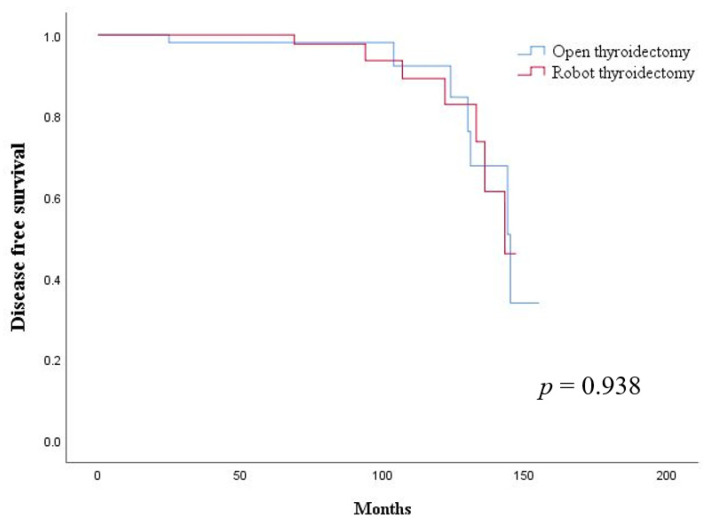
Disease-free survival between patients treated with robotic and open thyroidectomy.

**Table 1 cancers-13-03293-t001:** Clinical characteristics of patients with thyroid cancer treated with robotic versus open thyroidectomy.

Variable		Robotic (*n* = 99)	Open (*n* = 62)	*p*-Value
Sex, male:female		6:93	16:46	<0.001
Age (years)		16.9 ± 2.3 (8–19)	16.8 ± 3.0 (7–19)	0.559
BMI (kg/m^2^)		22.2 ± 3.9 (14.1–39.7)	22.0 ± 4.8 (13.2–35.9)	0.425
Diagnostic Sx				
	Incidentaloma	90 (90.9)	59 (95.2)	0.317
	Hypo- or Hyperthyroidism	3 (3.0)	1 (1.6)	0.551
	Others	6 (6.1)	2 (3.2)	0.551
Radiation exposure		0 (0.0)	0 (0.0)	
Family hx		20 (20.2)	11 (17.7)	0.700
	Thyroid cancer of first degree	12 (12.1)	2 (3.2)	
	Thyroid cancer of second degree	6 (6.1)	7 (11.3)	
	Others (Lung cancer, hypothyroidism)	2 (2.0)	2 (3.2)	

Data are expressed as the patient number (%) or mean ± SD (range). Statistically significant differences were defined as *p* < 0.05. Abbreviations: BMI, body mass index.

**Table 2 cancers-13-03293-t002:** Comparison of operation type and intraoperative outcomes between patients treated with robotic and open thyroidectomy.

Operation		Robotic (*n* = 99)	Open (*n* = 62)	*p*-Value
Operation type				
	Lobectomy	3 (3.0)	2 (3.2)	0.432
	Lobectomy with CCND	29 (29.3)	9 (14.5)	
	Ipsilateral TT and contralateral partial or subtotal thyroidectomy with CCND	14 (14.1)	3 (4.8)	
	BTT with CCND	30 (30.3)	15 (24.2)	0.010
	BTT with MRND	23 (23.2)	33 (53.2)	
Operation time (min)		171.2 ± 101.7 (69–635)	182.6 ± 98.2 (43–456)	0.496
Combined procedure during operation		3 (3.0)	7 (11.3)	0.035
	RLN shaving	3 (3.0)	1 (1.6)	
	Tracheal wall shaving	0 (0.0)	2 (3.2)	
	RLN and tracheal wall shaving	0 (0.0)	3 (4.8)	
	RLN shaving and tracheal re-anastomosis	0 (0.0)	1 (1.6)	

Data are expressed as the patient number (%) or mean ± SD (range). Statistically significant differences were defined as *p* < 0.05. Abbreviations: CCND, central compartment neck dissection; TT, total thyroidectomy; BTT, bilateral total thyroidectomy; MRND, modified radical neck dissection; RLN, recurrent laryngeal nerve.

**Table 3 cancers-13-03293-t003:** Comparison of pathologic findings between patients treated with robotic and open thyroidectomy.

Pathology		Robotic (*n* = 99)	Open (*n* = 62)	*p*-Value
Pathology of cancer				0.639
	PTC	94 (94.9)	58 (93.5)	
	FTC	4 (4.1)	4 (6.5)	
	PDTC	1 (1.0)	0 (0.0)	
Tumor size (cm)		1.8 ± 1.2 (0.2–5.3)	2.5 ± 1.8 (0.3–8.5)	0.010
Tumor number				0.134
	Single	70 (71.4)	34 (54.8)	
	Multiple	8 (8.2)	10 (16.1)	
	Bilateral	20 (20.4)	18 (29.0)	
Gross ETE positivity		48 (48.5)	39 (62.9)	0.165
No. of harvested LNs				
	Positive nodes of CCND	2.8 ± 3.2	6.0 ± 5.2	<0.001
	Total node of CCND	7.3 ± 5.1	11.3 ± 7.0	<0.001
	Positive node of MRND	6.2 ± 4.1	12.5 ± 11.2	0.012
	Total node of MRND	43.0 ± 31.2	59.3 ± 40.9	0.107

Data are expressed as the patient number (%) or mean ± SD (range). Statistically significant differences were defined as *p* < 0.05. Abbreviations: PTC, papillary thyroid cancer; FTC, follicular thyroid cancer; PDTC, poorly differentiated thyroid cancer; ETE, extrathyroidal extension; No., number; CCND, central compartment neck dissection; MRND, modified radical neck dissection.

**Table 4 cancers-13-03293-t004:** Comparison of TNM stage and BRAF mutation between patients treated with robotic and open thyroidectomy.

Stage		Robotic (*n* = 99)	Open (*n* = 62)	*p*-Value
T stage				0.010
	T 0	1 (1.0)	0 (0.0)	0.401
	T 1	38 (38.4)	11 (17.7)	
	T 2	12 (12.1)	7 (11.3)	
	T 3a	0 (0.0)	5 (8.1)	0.013
	T 3b	45 (45.5)	32 (51.6)	
	T 4a	3 (3.0)	7 (11.3)	
	T 4b	0 (0.0)	0 (0.0)	
N stage				0.001
	N 0	33 (33.3)	10 (16.1)	
	N 1a	43 (43.4)	19 (30.6)	
	N 1b	23 (23.2)	33 (53.2)	
M stage				0.009
	M 0	98 (99.0)	56 (90.3)	
	M 1	1 (1.0)	6 (9.7)	
BRAF mutation positivity		21 (50%) (*n* = 42)	18 (62.1%) (*n* = 29)	0.315

Data are expressed as the patient number (%) or mean ± SD (range). Statistically significant differences were defined as *p* < 0.05.

**Table 5 cancers-13-03293-t005:** Comparison of postoperative outcomes between patients treated with robotic and open thyroidectomy.

Characteristics		Robotic (*n* = 99)	Open (*n* = 62)	*p*-Value
Postoperative complication		21 (21.2)	25 (40.3)	0.009
	Transient hypocalcemia	15 (15.2)	18 (29.0)	0.076
	Permanent hypocalcemia	2 (2.0)	2 (3.2)	0.632
	Chyle leakage	2 (2.0)	3 (4.8)	0.316
	Infection	1 (1.0)	0 (0.0)	0.427
	Transient hoarseness	1 (1.0)	0 (0.0)	0.427
	RLN injury	0 (0.0)	1 (1.6)	0.205
	Horner’s syndrome	0 (0.0)	1 (1.6)	0.205
Length of hospital stay (days)		3.6 ± 1.1	4.8 ± 2.0	<0.001

Data are expressed as the patient number (%) or mean ± SD (range). Statistically significant differences were defined as *p* < 0.05. Abbreviations: RLN, recurrent laryngeal nerve.

**Table 6 cancers-13-03293-t006:** Comparison of follow-up time and recurrence rate and site between patients treated with robotic and open thyroidectomy.

Recurrence		Robotic (*n* = 99)	Open (*n* = 62)	*p*-Value
Follow-up time (months)		67.0 ± 38.5 (6–147)	72.1 ± 42.7 (8–155)	0.569
Recurrence rate		7 (7.1)	7 (11.3)	0.355
Recurrence site				0.472
	Local	3 (3.0)	5 (8.1)	
	Distant	3 (3.0)	1 (1.6)	
	Local and distant	1 (1.0)	1 (1.6)	

Data are expressed as the patient number (%) or mean ± SD (range). Statistically significant difference was defined as *p* < 0.05.

## Data Availability

The data presented in this study are available on request from Severance Hospital. The data are not publicly available owing to the Personal Information Protection Act.

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
