# Peer review of "Comparison of Surgical Outcomes between Robotic Transaxillary and Conventional Open Thyroidectomy in Pediatric Thyroid Cancer"

_cancers, 2021, doi:10.3390/cancers13133293_

Round 1

Reviewer 1 Report

The paper assessed 161 pediatric patients (≤ 19 years of age) at the time of diagnosis who underwent robotic trans-axillary thyroidectomy or conventional open thyroidectomy due to thyroid cancer. Data presentation, analysis and conclusions are generally coherent. The authors described the procedure, underlying that pediatric thyroid cancer has been increasing steadily over the past 30 years and trans-axillary thyroidectomy serves as a minimally invasive surgical alternative to conventional open thyroidectomy. The advantage of the paper is a large cohort of pediatric patients undergoing analysis. The paper is well written.

However, I have some doubts, which I would like to address.

The authors reported that the patients chose themselves the surgical procedure. Firstly, it seems to me that the decision was rather taken by their parents or guardians at least in some patients. The paper does not specify the percentage of patients over 18 years of age. If so, some decisions were then taken by the patients themselves or their parents/legal guardians.

However, my greater concern is related that the fact that carcinoma was more advanced in patients who underwent open thyroidectomy as assessed by the TNM. It also seems that during qualification, it was the surgeon who recommended the procedure, choosing a classic approach in more advanced disease stages. Please, explain it.  

It seems to me that trans-axillary thyroidectomy should be reserved to less advanced cases, although the authors in the Discussion section stressed that it is also possible and safe in advanced cases with a tumor size higher than 4 cm with the presence of ETE, which is clearly shown by the presented material.  

The authors rightly noticed that the study limitation includes a selective bias of more advanced cancers in the COT group, which makes it difficult to establish the better efficacy of trans-axillary thyroidectomy compared to conventional open thyroidectomy for highly advanced thyroid cancer.

I agree that trans-axillary thyroidectomy is connected to the opportunity to improve the cosmetic effects. However, I find it difficult to agree with the improvement in surgical outcomes, which are similar to open thyroidectomy. Please rephrase.

Thank you.

Author Response

Dear editor.

Thank you for pointing out the specific improvement.

And, I have summerized and answered what you said below.

  1. Decision of surgical procedure

As you mentioned, these patients were pediatric population. So I revised the patients chose to the patients and parents’ choice.

  1. Many advanced cases in COT

As mentioned in lines 267-270, this disproportionate distribution may be explained by the social perception that open surgery is more accurate for highly advanced cancer or the surgeon’s recommendation at the outpatient clinic.

  1. Improvement in surgical outcomes

I totally agree with your opinion that the improvement of surgical outcomes of RT is difficult to establish the better efficacy to COT because of selective bias. So, I erased the “or better” word in line 298, and I revised the sentence in line 328-333 to “Second, although it is difficult to accurately compare efficiency between the procedures because the number of advanced cancer cases was higher in the open group than in robotic group, RT was associated with a similar operation time, recurrence rate, and DFS as COT, and yielded better results, with respect to a reduced length of hospital stay and low risk of postoperative complications, than COT.”

In addition, the limitation of this study in line 336 mentioned this selective bias makes difficult to clearly elucidate the superiority of RT to COT for highly advanced thyroid cancer. So in line 340, I recommend prospective, multicenter controlled studies are needed.

Please refer to the attached revised manuscript.

Thank you again.

Best regards

In A, Lee

Reviewer 2 Report

Lee et al. reported a retrospective study including 161 pediatric patients who underwent robotic trans-axillary or conventional open thyroidectomy. Robotic thyroidectomy appears to be a feasible and safe option for pediatric patients.

The topic is quite interesting and conclusions are both clear and adequately supported by results.

Several comments are suggested to improve on the submitted manuscript:

  • The design of this study should be better described (number and experience of leader surgeons, criteria adopted for selection of robotic or open procedure, etc).
  • Authors reported that: “patients chose their preferred surgical procedure”. However, the mean tumor size in patients with open procedure was significantly higher that than reported in patients with robotic thyroidectomy. Is it possible to hypothesize an influence from the surgeon on the selection of the procedure?
  • The treatment of differentiated thyroid cancer has been changing. In low and intermediate risk differentiated thyroid cancer, surgery is becoming more conservative. This point should be reported in the introduction. Please, see and cite: PMID: 32623845
  • The authors should better develop the future perspectives of this study.
  • Please, report a brief description of other procedures to minimize perioperative complications (intraoperative neuromonitoring, etc). Please, see and cite: PMID: 29546741

Author Response

Dear editor.

Thank you for pointing out the specific improvement.

And, I have summerized and answered what you said below.

  1. The design of the study with case selection

I have added these sentences at your recommendation;

In line 74, “At out institution, eight surgeons were involved in this study.”

In line 77, “Six surgeons were experts of RT, and two doctors performed only COT. In some high advanced cases, COT was recommended as per the surgeon's opinion.”

  1. the treatment of DTC

I have added these sentences at your recommendation;

In line 240, “Moreover, surgical treatment has been shifting toward a conservative approach in patients with low and intermediate risk differentiated thyroid cancer in recent years” with citation.

  1. perioperative complication

I have added these sentences at your recommendation;

In line 277, “To minimize the rate of intraoperative complications, additionally our institution introduced intraoperative neuromonitoring in some cases, such as clinical N1, clinically suspicious T4, and preoperative vocal cord paralysis, from 2015 and no cases of postoperative RLN injury were reported between 2015 and 2018.” with citation.

Please refer to the attached revised manuscript.

Thank you again.

Best regards

In A, Lee
